# Global Stability Analysis of a Bioreactor Model for Phenol and Cresol Mixture Degradation

**Neli Dimitrova** [1,*,†] **and Plamena Zlateva** [1,2,†]

[1] Institute of Mathematics and Informatics, Bulgarian Academy of Sciences, Acad. G. Bonchev Str. Block 8, 1113 Sofia, Bulgaria
[2] Institute of Robotics, Bulgarian Academy of Sciences, Acad. G. Bonchev Str. Block 2, 1113 Sofia, Bulgaria; plamzlateva@abv.bg
[*] Correspondence: nelid@math.bas.bg
[†] These authors contributed equally to this work.

**Abstract:** We propose a mathematical model for phenol and $p$-cresol mixture degradation in a continuously stirred bioreactor. The model is described by three nonlinear ordinary differential equations. The novel idea in the model design is the biomass specific growth rate, known as sum kinetics with interaction parameters (SKIP) and involving inhibition effects. We determine the equilibrium points of the model and study their local asymptotic stability and bifurcations with respect to a practically important parameter. Existence and uniqueness of positive solutions are proved. Global stabilizability of the model dynamics towards equilibrium points is established. The dynamic behavior of the solutions is demonstrated on some numerical examples.

**Keywords:** mathematical model; continuous bioreactor; biodegradation; phenol and $p$-cresol mixture; SKIP model; equilibrium points; stability analysis; global stabilizability; numerical simulation



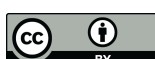

## 1. Introduction

Organic chemical mixtures are among the most persistent environmental pollutants. Different aromatic compounds such as phenol, cresols, nitrophenols, benzene, etc. coexist as complex mixtures in wastewaters from petroleum refineries, coal mining and other industrial chemical sources [1]. Biological degradation has recently become a viable technology for remediation of organic pollutants as an alternative to the traditional physical and chemical methods that can be costly and produce hazardous products. Most of the current research has been directed to the isolation and study of microbial species with high-degradation activity and capabilities of degrading chemical compounds. The review paper [2] reports on hundreds of isolated bacteria capable of degrading aromatic compounds, among them different strains of *Aspergillus awamori*, *Arthrobacter*, *Burkholderia*, *Mycobacterium*, *Pseudomonas*, *Rhodococcus*, *Staphylococcus*, *Trametes hirsute* etc. The biodegradation of one or all chemical components depends on the composition of the particular mixture and the used microorganisms [3–5]. The adequate analysis of interactions between the compounds and their influence on microbial growth is very important for understanding the simultaneous metabolism of phenolic mixtures [6].

Most research on microbial potentials to degrade chemical pollutants has been performed on a laboratory scale. Based on batch processes various mathematical biodegradation kinetic models have been recently developed and widely used. Among them are Monod's, Haldane's (known also as Andrews), sum kinetic models, sum kinetics with interaction parameter (SKIP) models, etc. [7,8]. It is known that Monod's and Haldane's models are appropriate for single substrate utilization. The Monod model describes the biodegradation rate in dependence of the biomass concentration. When a substrate inhibits its own degradation then Haldane's model is more appropriate. In [9] the Haldane equation modified with a Monod-like switching function is proposed and applied to the biological

removal of mixtures of phenolic compounds in sequential batch bioreactors. In [10] the aerobic biodegradability of phenol, resorcinol and 5-methylresorcinol and their different two-component mixtures is investigated and various kinetic models are tested to obtain the best curve fit.

In the case when a mixture of two or more substrates occurs, the sum kinetic and SKIP models predict better the outcome of biodegradation experiments. The latter have been proposed for the first time in [11] and widely used by many researchers. The (no-interaction) sum kinetics model for cell growth is usually represented as a sum of the specific growth rates on each substrate, e.g., as a sum of Monod- and/or Haldane-type specific growth rates. These models were evaluated in [12,13] for biodegradation of benzene, toluene and phenol mixtures using *Pseudomonas putida* F1 and *Burkholderia* sp. strain JS150 and found that the interactions between these substrates could not be described by sum kinetics models. On the contrary, the SKIP model predicts better the outcome of the mixed-culture experiments. This is due the fact that the SKIP models extend the sum kinetics models by incorporating interaction parameters to describe more accurately the biodegradation of the chemical mixture.

The biodegradation of benzene, toluene and phenol is studied in [14] by adaptation of *Pseudomonas putida* F1 ATCC 700007. For the substrate mixtures, a SKIP model is used. The latter provides an excellent prediction of the growth kinetics and the interactions between these substrates.

In [15] biodegradation kinetics of different multiple substrate mixtures of mono-aromatic volatile organic carbon (VOCs) such as toluene, ethyl benzene and o-xylene are studied. A general mixed-substrate biodegradation model is developed which can describe the biodegradation kinetics of common industrial VOCs when present as a mixture, incorporating parameters for interaction effects.

The paper [16] examines biodegradation kinetics of styrene and ethylbenzene, independently and as binary mixtures, using a series of aerobic batch degradation. The SKIP model and the purely competitive enzyme kinetics model are employed to evaluate any interactions. The SKIP model is found to more accurately describe the interactions.

Here, we propose a mathematical model for biodegradation of phenol and 4-methylphenol (*p*-cresol) in a *continuously stirred tank bioreactor*, in which the biodegradation kinetics is described by a SKIP model. The bioreactor model presents an extension of the growth kinetic model proposed in [17]. There, the growth behavior and degradation capacity of *Aspergillus awamori* NRRL 3112 microbial strain on the binary mixture phenol/*p*-cresol are investigated. Based on laboratory experiments, the growth kinetic model is first evaluated by a sum kinetic model involving Haldane's specific growth rate. An alternative model is then formulated by adding interaction parameters into the sum kinetics model to produce the SKIP model. It is shown that the SKIP model describes better the degradation patterns in the biological system.

The paper is organized as follows. The next Section 2 presents a short description of the proposed mathematical model. Section 3 includes steady states computations. Local stability analysis and bifurcations of the equilibrium points are presented in Section 4. Section 5 reports on general and important properties of the model solutions and provides results on the global stabilizability of the system towards an interior equilibrium point. The last Section 6 presents numerical examples as illustration of the theoretical studies on the model dynamics.

## 2. Model Description

We consider the following mathematical model for phenol and *p*-cresol mixture degradation in a continuously stirred bioreactor

$$\frac{dX(t)}{dt} = \left(\mu(S_{ph}, S_{cr}) - D\right)X(t) \tag{1}$$

$$\frac{dS_{ph}(t)}{dt} = -k_{ph}\,\mu(S_{ph}, S_{cr})X(t) + D(S_{ph}^0 - S_{ph}(t)) \tag{2}$$

$$\frac{dS_{cr}(t)}{dt} = -k_{cr}\,\mu(S_{ph}, S_{cr})X(t) + D(S_{cr}^0 - S_{cr}(t)), \tag{3}$$

where $\mu(S_{ph}, S_{cr})$ is the specific growth rate, presented by

$$\mu(S_{ph}, S_{cr}) = \frac{\mu_{max(ph)}S_{ph}}{k_{s(ph)} + S_{ph} + \frac{S_{ph}^2}{k_{i(ph)}} + I_{cr/ph}S_{cr}} + \frac{\mu_{max(cr)}S_{cr}}{k_{s(cr)} + S_{cr} + \frac{S_{cr}^2}{k_{i(cr)}} + I_{ph/cr}S_{ph}}. \tag{4}$$

The definition of the state variables $X$, $S_{ph}$ and $S_{cr}$ as well as of the model parameters is given in Table 1. The numerical values in the last column are validated by laboratory experiments and given in [17].

The specific growth rate $\mu(S_{ph}, S_{cr})$ represents a SKIP (sum kinetics with interaction parameters) model for biological degradation of the chemical compounds. The interaction parameters $I_{cr/ph}$ and $I_{ph/cr}$ indicate the degree to which substrate *p*-cresol affects the biodegradation of substrate phenol, and substrate phenol affects the biodegradation of substrate *p*-cresol, respectively. The larger value of $I_{cr/ph}$ (see Table 1) indicates that *p*-cresol inhibits the utilization of phenol much more than phenol inhibits the utilization of *p*-cresol. The kinetic function $\mu(S_{ph}, S_{cr})$ also involves inhibition terms $\frac{S_{ph}^2}{k_{i(ph)}}$ and $\frac{S_{cr}^2}{k_{i(cr)}}$ for cell growth on phenol and *p*-cresol, respectively. Obviously, $\mu(S_{ph}, 0)$ and $\mu(0, S_{cr})$ are the well-known Andrews (or Haldane) model functions, which are unimodal and achieve their maximum at $S_{ph} = \sqrt{k_{s(ph)}k_{i(ph)}}$ and $S_{cr} = \sqrt{k_{s(cr)}k_{i(cr)}}$ respectively.

The influent concentrations $S_{ph}^0$, $S_{cr}^0$ and the dilution rate $D$ are the parameters that can be manipulated by the operator of the bioreactor. In our analysis we assume that $S_{ph}^0$ and $S_{cr}^0$ are constant and consider the dilution rate $D$ as a varying control parameter. Clearly, $D > 0$ should be fulfilled.

The same model (1)–(3) has been considered in [18] using a more simple specific growth rate function $\mu(S_{ph}, S_{cr})$ which does not involve the inhibition terms $\frac{S_{ph}^2}{k_{i(ph)}}$ and $\frac{S_{cr}^2}{k_{i(cr)}}$ for cell growth on phenol and *p*-cresol. Adding these terms makes the dynamics (1)–(3) more complicated, but as shown in [17], see also [5], the SKIP model (4) describes the trend of experimental data much better than other kinetic models.

**Table 1.** Model variables and parameters.

| | Definitions | Values |
|---|---|---|
| $X$ | biomass concentration $[\text{g}/\text{dm}^3]$ | – |
| $S_{ph}$ | phenol concentration $[\text{g}/\text{dm}^3]$ | – |
| $S_{cr}$ | $p$-cresol concentration $[\text{g}/\text{dm}^3]$ | – |
| $D$ | dilution rate $[h^{-1}]$ | – |
| $S_{ph}^0$ | influent phenol concentration $[\text{g}/\text{dm}^3]$ | 0.7 |
| $S_{cr}^0$ | influent $p$-cresol concentration $[\text{g}/\text{dm}^3]$ | 0.3 |
| $k_{ph}$ | metabolic coefficient $[S_{ph}/X]$ | 11.7 |
| $k_{cr}$ | metabolic coefficient $[S_{cr}/X]$ | 5.8 |
| $k_{i(ph)}$ | inhibition constant for cell growth on phenol $[\text{g}/\text{dm}^3]$ | 0.61 |
| $k_{i(cr)}$ | inhibition constant for cell growth on cresol $[\text{g}/\text{dm}^3]$ | 0.45 |
| $I_{ph/cr}$ | interaction coefficient indicating the degree to which phenol affects the $p$-cresol biodegradation | 0.3 |
| $I_{cr/ph}$ | interaction coefficient indicating the degree to which $p$-cresol affects the phenol biodegradation | 8.6 |
| $\mu_{max(ph)}$ | maximum specific growth rate on phenol as a single substrate $[h^{-1}]$ | 0.23 |
| $\mu_{max(cr)}$ | maximum specific growth rate on $p$-cresol as a single substrate $[h^{-1}]$ | 0.17 |
| $k_{s(ph)}$ | saturation constant for cell growth on phenol $[\text{g}/\text{dm}^3]$ | 0.11 |
| $k_{s(cr)}$ | saturation constant for cell growth on $p$-cresol $[\text{g}/\text{dm}^3]$ | 0.35 |

## 3. Existence of Equilibrium Points

We shall investigate existence of the model equilibrium points in dependence of the control parameter $D$.

The equilibrium points of (1)–(3) are solutions of the following system of algebraic equations

$$\left(\mu(S_{ph}, S_{cr}) - D\right)X = 0 \tag{5}$$

$$-k_{ph}\,\mu(S_{ph}, S_{cr})X + D(S_{ph}^0 - S_{ph}) = 0 \tag{6}$$

$$-k_{cr}\,\mu(S_{ph}, S_{cr})X + D(S_{cr}^0 - S_{cr}) = 0. \tag{7}$$

Obviously, the point $E_0 = (0, S_{ph}^0, S_{cr}^0)$ (with $X = 0$) is an equilibrium point of the model for all $D > 0$.

We are looking now for solutions of (5)–(7) assuming that $X \not\equiv 0$.

After multiplying Equation (6) by $-k_{cr}$, Equation (7) by $k_{ph}$ and summing the latter, we obtain

$$-k_{cr}(S_{ph}^0 - S_{ph}) + k_{ph}(S_{cr}^0 - S_{cr}) = 0. \tag{8}$$

Let us express $S_{ph}$ from (8) as a function of $S_{cr}$. Denoting

$$K = \frac{k_{ph}}{k_{cr}}, \quad S^0 = S_{ph}^0 - KS_{cr}^0, \tag{9}$$

We obtain

$$S_{ph} = S_{ph}^0 - \frac{k_{ph}}{k_{cr}}\left(S_{cr}^0 - S_{cr}\right) = S^0 + KS_{cr}. \tag{10}$$

After replacing the latter presentation of $S_{ph}$ into the equation $\mu(S_{ph}, S_{cr}) = D$ from (5) we obtain an equation with respect to $S_{cr}$ of the form

$$\mu(S^0 + KS_{cr}, S_{cr}) = D,$$

or equivalently

$$\frac{\mu_{max(ph)}\left(S^0 + KS_{cr}\right)}{k_{s(ph)} + S^0 + KS_{cr} + \frac{1}{k_{i(ph)}}(S^0 + KS_{cr})^2 + I_{cr/ph}S_{cr}}$$
$$+ \frac{\mu_{max(cr)}S_{cr}}{k_{s(cr)} + S_{cr} + \frac{1}{k_{i(cr)}}S_{cr}^2 + I_{ph/cr}(S^0 + KS_{cr})} = D.$$

Straightforward calculations lead to a polynomial equation of the form

$$A_1 S_{cr}^4 + A_2 S_{cr}^3 + A_3 S_{cr}^2 + A_4 S_{cr} + A_5 = 0, \tag{11}$$

where

$$A_1 = -D \cdot \frac{1}{k_{i(cr)}} \cdot \frac{1}{k_{i(ph)}};$$

$$A_2 = \mu_{max(ph)}\frac{K}{k_{i(cr)}} + \mu_{max(cr)}\frac{1}{k_{i(ph)}}$$
$$- D\left[(1 + I_{ph/cr}K)\frac{1}{k_{i(ph)}} + \frac{1}{k_{i(cr)}}\left(I_{cr/ph} + K + 2KS^0\frac{1}{k_{i(ph)}}\right)\right];$$

$$A_3 = \mu_{max(ph)}\left[S^0\frac{1}{k_{i(cr)}} + K(1 + I_{ph/cr}K)\right] + \mu_{max(cr)}\left[K + I_{cr/ph} + 2KS^0\frac{1}{k_{i(ph)}}\right]$$
$$- D\left[\frac{1}{k_{i(ph)}}(k_{s(cr)} + I_{ph/cr}S^0) + (1 + I_{ph/cr}K)\left(K + I_{cr/ph} + 2KS^0\frac{1}{k_{i(ph)}}\right)\right.$$
$$\left. + \frac{1}{k_{i(cr)}}\left(k_{s(ph)} + S^0 + \frac{1}{k_{i(ph)}}S^{0\,2}\right)\right];$$

$$A_4 = \mu_{max(ph)}\left[S^0(1 + I_{ph/cr}K) + K(k_{s(cr)} + I_{ph/cr}S^0)\right]$$
$$+ \mu_{max(cr)}\left(k_{s(ph)} + S^0 + \frac{1}{k_{i(ph)}}S^{0\,2}\right)$$
$$- D\left[(k_{s(cr)} + I_{ph/cr}S^0)\left(K + I_{cr/ph} + 2KS^0\frac{1}{k_{i(ph)}}\right)\right.$$
$$\left. + (1 + I_{ph/cr}K)\left(k_{s(ph)} + S^0 + \frac{1}{k_{i(ph)}}S^{0\,2}\right)\right];$$

$$A_5 = \left[\mu_{max(ph)}S^0 - D\left(k_{s(ph)} + S^0 + \frac{1}{k_{i(ph)}}S^{0\,2}\right)\right](k_{s(cr)} + I_{ph/cr}S^0).$$

All coefficients $A_i$, $i = 1, 2, \ldots, 5$, depend on the parameter $D$.

Obviously, if $A_5 = 0$, then Equation (11) possesses a solution $S_{cr} = 0$. We have

$$A_5 = 0 \iff D = D_{cr} := \frac{S^0\mu_{max(ph)}k_{i(ph)}}{S^{0\,2} + k_{i(ph)}S^0 + k_{i(ph)}k_{s(ph)}} = \mu(S^0, 0). \tag{12}$$

The latter value of $D_{cr}$ is biologically reasonable only if $S^0 > 0$. Using the numerical values of the model coefficients in the last column of Table 1, we obtain

$$S^0 = S_{ph}^0 - KS_{cr}^0 \approx 0.09483 > 0,$$

and so,

$$D_{cr} = \mu(S^0, 0) \approx 0.09933.$$

This means that for $D = D_{cr}$ there exists an equilibrium point with $S_{cr} = 0$. Further from (10) we compute the component of $S_{ph} = S^0$, and from (7) we get the corresponding component of $X = \dfrac{S_{cr}^0}{k_{cr}}$. Thus, at $D = D_{cr}$ there exists a steady state

$$E_1 = E_1(D_{cr}) = \left( \frac{S_{cr}^0}{k_{cr}}, S^0, 0 \right) = (0.05172, 0.09483, 0). \tag{13}$$

Considering the cubic equation $A_1 S_{cr}^3 + A_2 S_{cr}^2 + A_3 S_{cr} + A_4 = 0$ at $D = D_{cr}$ (i.e., with $A_5 = 0$), numerical computations produce the following roots of the latter equation

$$-4.484933737, \quad 0.2614282531 \pm i\, 0.2468184467,$$

so, the real root is negative and cannot serve as a component of the model equilibrium point.

If $D \neq D_{cr}$ then Equation (11) may possess up to 4 real positive solutions with respect to $S_{cr}$. If there exists at least one positive solution of (11), say $S_{cr}^*$, such that $S_{cr}^* < S_{cr}^0$ for some values of $D$, we shall have an interior (with positive components) equilibrium of the form

$$E^* = (X^*, S_{ph}^*, S_{cr}^*), \quad S_{ph}^* = S^0 + K S_{cr}^* < S_{ph}^0, \quad X^* = \frac{S_{cr}^0 - S_{cr}^*}{k_{cr}} = \frac{S_{ph}^0 - S_{ph}^*}{k_{ph}}. \tag{14}$$

**Remark 1.** *If we express $S_{cr}$ from (8) as a function of $S_{ph}$ and denote $\hat{K} = \dfrac{k_{cr}}{k_{ph}} = \dfrac{1}{K}$, $\hat{S}^0 = S_{cr}^0 - \hat{K} S_{ph}^0$, then we shall have $S_{cr} = \hat{S}^0 + \hat{K} S_{ph}$. Similar calculations as above will produce a polynomial equation of the form $\hat{A}_1 S_{ph}^4 + \hat{A}_2 S_{ph}^3 + \hat{A}_3 S_{ph}^2 + \hat{A}_4 S_{ph} + \hat{A}_5 = 0$, where the coefficients $\hat{A}_i$ are similar to $A_i$, $i = 1, 2, \ldots, 5$, within $\hat{S}^0$ and $\hat{K}$ instead of $S^0$ and $K$, respectively. In this case we have*

$$\hat{A}_5 = \left( \mu_{max,cr} - D \left( k_{s(cr)} + \hat{S}^0 + \frac{\hat{S}^{0^2}}{k_{i(cr)}} \right) \right) (k_{s(ph)} + I_{cr/ph} \hat{S}^0).$$

*Obviously, $\hat{A}_5 = 0$ at $\hat{D} = \mu(0, \hat{S}^0)$. But in this case $\hat{S}^0 = -\frac{1}{K} S^0 \approx -0.047 < 0$, thus there is no value of $D$ at which $S_{ph} = 0$ is a root of the polynomial $\sum_{i=1}^{5} \hat{A}_i S_{ph}^{5-i} = 0$. As we shall see in the following, this is the case with the equilibrium component $S_{ph}$.*

Numerical computations show that if $D > D_{cr}$ then there are no positive real roots of Equation (11). Therefore, we can expect interior (coexistence) equilibria of the form $E^*$ if $D \in (0, D_{cr})$, in case that the equilibrium components with respect to $S_{cr}$ satisfy the inequality $S_{cr} \leq S_{cr}^0$. Further we obtain numerically the following results:

- There exists a value $D = D_{cr}^{(1)} \approx 0.0745599$, so that Equation (11) possesses a double root $S_{cr} \approx 0.04327$ for $D = D_{cr}^{(1)}$.

- If $D < D_{cr}^{(1)}$ then there are no positive roots of (11) which are less than or equal to $S_{cr}^0$.

- Denote $D_{cr}^{(2)} := \mu(S_{ph}^0, S_{cr}^0) \approx 0.08651 < D_{cr}$. If $D \in \left( D_{cr}^{(1)}, D_{cr}^{(2)} \right)$ then there are two positive roots of (11) which are less than $S_{cr}^0$.

- If $D \in \left( D_{cr}^{(2)}, D_{cr} \right)$, $D_{cr} = \mu(S^0, 0) \approx 0.09933$, then there is only one positive root of (11) which is less than $S_{cr}^0$.

The left plot in Figure 1 shows the graph of the function $\mu(S^0 + KS_{cr}, S_{cr})$ for $S_{cr} \in [0, S^0_{cr}] = [0, 0.3]$; the horizontal dash lines correspond to the values of $D^{(1)}_{cr}$, $D^{(2)}_{cr}$ and $D_{cr}$.

Therefore, the model (1)–(3) possesses two interior equilibrium points depending on the values of $D$. Denote them by

$$
\begin{aligned}
E_2 = E_2(D) &= \left(X^{(2)}, S^{(2)}_{ph}, S^{(2)}_{cr}\right), \quad D \in \left(D^{(1)}_{cr}, D_{cr}\right); \\
E_3 = E_3(D) &= \left(X^{(3)}, S^{(3)}_{ph}, S^{(3)}_{cr}\right), \quad D \in \left(D^{(1)}_{cr}, D^{(2)}_{cr}\right), \quad \text{with } S^{(3)}_{cr} > S^{(2)}_{cr}.
\end{aligned}
$$

Numerical computations also produce the following results:

$$E_2(D_{cr}) = E_1 = (0.05172, \ 0.09483, \ 0), \qquad\qquad D_{cr} = \mu(S^0, 0) = 0.09933;$$

$$E_2(D^{(1)}_{cr}) = E_3(D^{(1)}_{cr}) = (0.04426, 0.18211, 0.04327), \quad D^{(1)}_{cr} = 0.0745599;$$

$$E_3(D^{(2)}_{cr}) = E_0 = (0, S^0_{ph}, S^0_{cr}) = (0, 0.7, 0.3), \qquad D^{(2)}_{cr} = \mu(S^0_{ph}, S^0_{cr}) = 0.08651.$$

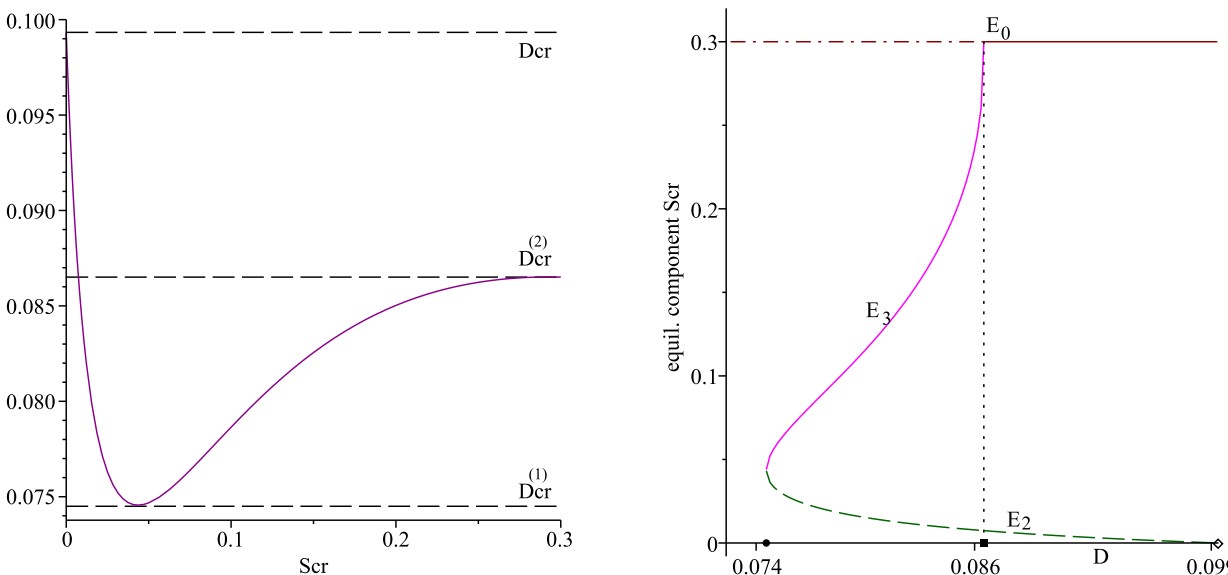

**Figure 1. (Left)**: graph of the function $\mu(S^0 + KS_{cr}, S_{cr})$ for $S_{cr} \in [0, S^0_{cr}]$. **(Right)**: the equilibrium components $S^{(2)}_{cr}$ (dash line) and $S^{(3)}_{cr}$ (solid line), parameterized on $D$. The horizontal dash-dot&solid line passes trough $S^0_{cr}$. On the horizontal axis, the solid circle denotes $D^{(1)}_{cr}$, the solid box denotes $D^{(2)}_{cr}$, the diamond denotes $D_{cr}$. The vertical dot line passes through $D^{(2)}_{cr}$.

Figure 1 (right plot) and Figure 2 visualize the components $S_{cr}$, $S_{ph}$ and $X$ of the equilibria $E_0$, $E_2$ and $E_3$. In the three plots, the components of the equilibrium point $E_0$ are marked by horizontal dash-dot&solid lines, the components of $E_2$ are marked by dash lines and the ones of $E_3$ are shown by solid lines.

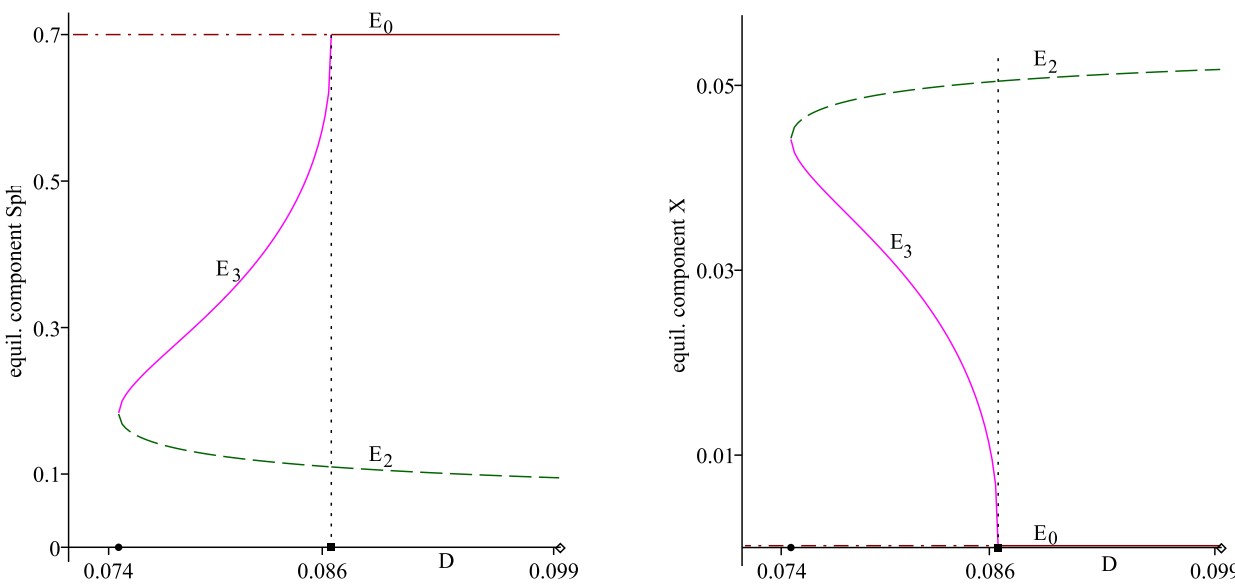

**Figure 2.** (**Left**): the equilibrium components $S_{ph}^{(2)}$ (dash line) and $S_{ph}^{(3)}$ (solid line), parameterized on $D$. The horizontal dash-dot&solid line passes trough $S_{ph}^0$. (**Right**): the equilibrium components $X^{(2)}$ (dash line) and $X^{(3)}$ (solid line), parameterized on $D$. The horizontal dash-dot&solid line passes trough 0. On the horizontal axis (left and right plot), the solid circle denotes $D_{cr}^{(1)}$, the solid box denotes $D_{cr}^{(2)}$, the diamond denotes $D_{cr}$. The vertical dot line passes through $D_{cr}^{(2)}$.

## 4. Local Stability of the Equilibrium Points

In this section we shall study the conditions for local asymptotic stability of the model equilibrium points.

It is well known that an equilibrium point is locally asymptotically stable, if all eigenvalues of the Jacobi matrix evaluated at this equilibrium have negative real parts, cf. e.g., [19]. The eigenvalues of the Jacobi matrix coincide with the roots of the corresponding characteristic polynomial.

To simplify notations, in the following we shall sometimes write $\mu$ instead of $\mu(S_{ph}, S_{cr})$. The Jacobi matrix $J$ related to the model Equations (1)–(3) has the form

$$
J = \begin{pmatrix}
\mu(S_{ph}, S_{cr}) - D & \frac{\partial \mu}{\partial S_{ph}} X & \frac{\partial \mu}{\partial S_{cr}} X \\
-k_{ph}\mu(S_{ph}, S_{cr}) & -k_{ph}\frac{\partial \mu}{\partial S_{ph}} X - D & -k_{ph}\frac{\partial \mu}{\partial S_{cr}} X \\
-k_{cr}\mu(S_{ph}, S_{cr}) & -k_{cr}\frac{\partial \mu}{\partial S_{ph}} X & -k_{cr}\frac{\partial \mu}{\partial S_{cr}} X - D
\end{pmatrix}.
$$

The characteristic polynomial corresponding to $J$ is defined by $det(J - \lambda I_3)$, where $\lambda$ is any complex number and $I_3$ is the $(3 \times 3)$–identity matrix

$$
det(J - \lambda I_3) = \begin{vmatrix}
\mu(S_{ph}, S_{cr}) - D - \lambda & \frac{\partial \mu}{\partial S_{ph}} X & \frac{\partial \mu}{\partial S_{cr}} X \\
-k_{ph}\mu(S_{ph}, S_{cr}) & -k_{ph}\frac{\partial \mu}{\partial S_{ph}} X - D - \lambda & -k_{ph}\frac{\partial \mu}{\partial S_{cr}} X \\
-k_{cr}\mu(S_{ph}, S_{cr}) & -k_{cr}\frac{\partial \mu}{\partial S_{ph}} X & -k_{cr}\frac{\partial \mu}{\partial S_{cr}} X - D - \lambda
\end{vmatrix}.
$$

Multiplying the second row of the above determinant by $-\dfrac{k_{cr}}{k_{ph}}$ and adding the latter to the third row, we obtain

$$det(J - \lambda I_3) = \begin{vmatrix} \mu(S_{ph}, S_{cr}) - D - \lambda & \frac{\partial \mu}{\partial S_{ph}} X & \frac{\partial \mu}{\partial S_{cr}} X \\ -k_{ph}\mu(S_{ph}, S_{cr}) & -k_{ph}\frac{\partial \mu}{\partial S_{ph}} X - D - \lambda & -k_{ph}\frac{\partial \mu}{\partial S_{cr}} X \\ 0 & \frac{k_{cr}}{k_{ph}}(D + \lambda) & -D - \lambda \end{vmatrix}.$$

Straightforward calculations deliver the following characteristic polynomial

$$det(J - \lambda I_3) = (D + \lambda)^2 \left[ \mu(S_{ph}, S_{cr}) - D - \lambda - X \left( k_{ph}\frac{\partial \mu}{\partial S_{ph}} + k_{cr}\frac{\partial \mu}{\partial S_{cr}} \right) \right]. \quad (15)$$

Denote by $J(E_i)$ the Jacobian matrix evaluated at the equilibrium point $E_i$, $i = 0, 1, 2, 3$. It follows from (15) that $\lambda_{1,2} = -D < 0$ are always eigenvalues of $J(E_i)$, $i = 0, 1, 2, 3$. The third eigenvalue $\lambda_3$ is determined from the second multiplier of (15).

**Proposition 1.**

(i)   If $D < D_{cr}^{(2)} = \mu(S_{ph}^0, S_{cr}^0)$ then the equilibrium point $E_0 = \left( 0, S_{ph}^0, S_{cr}^0 \right)$ (with $X = 0$) is locally asymptotically unstable (a saddle).

(ii)   If $D > D_{cr}^{(2)}$ then $E_0$ is locally asymptotically stable (a stable node).

(iii)   At $D = D_{cr}^{(2)}$ the equilibrium $E_0$ is neither stable, nor unstable: $J(E_0)$ possesses a zero eigenvalue, $\lambda_3 = 0$, thus $D_{cr}^{(2)}$ is a bifurcation parameter value.

(iv)   The equilibrium point $E_1 = E_1(D_{cr}) = \left( \frac{S_{cr}^0}{k_{cr}}, S^0, 0 \right)$, (see (13)), is locally asymptotically unstable.

**Proof.** $(i)$–$(iii)$ We obtain from (15)

$$det(J(E_0) - \lambda I_3) = (D + \lambda)^2 (\mu(S_{ph}^0, S_{cr}^0) - D - \lambda),$$

thus the third root $\lambda_3$ satisfies

$$\lambda_3 = \mu(S_{ph}^0, S_{cr}^0) - D \begin{cases} > 0, & \text{if } D < D_{cr}^{(2)} = \mu(S_{ph}^0, S_{cr}^0), \\ < 0, & \text{if } D > D_{cr}^{(2)}, \\ = 0, & \text{if } D = D_{cr}^{(2)}. \end{cases}$$

$(iv)$ The characteristic polynomial corresponding to the equilibrium $E_1$ is presented by

$$det(J(E_1) - \lambda I_3) = -(D_{cr} + \lambda)^2 \left( S_{cr}^0 \left( K\frac{\partial \mu}{\partial S_{ph}}(S^0, 0) + \frac{\partial \mu}{\partial S_{cr}}(S^0, 0) \right) + \lambda \right).$$

The third root $\lambda_3$ of the latter polynomial is computed numerically and is equal to

$$\lambda_3 = -S_{cr}^0 \left( K\frac{\partial \mu}{\partial S_{ph}}(S^0, 0) + \frac{\partial \mu}{\partial S_{cr}}(S^0, 0) \right) \approx -(-0.7574) > 0,$$

which means that $E_1(D_{cr})$ is a saddle equilibrium point. This proves the proposition.  □

The equilibrium components $S_{ph}^{(i)}$ and $S_{cr}^{(i)}$ of the equilibria $E_i$, $i = 2, 3$, satisfy the equation $\mu(S_{ph}, S_{cr}) = D$, so that from (15) we obtain

$$det(J(E_i) - \lambda I_3) = -(D + \lambda)^2 \left[ \lambda + X^{(i)} \left( k_{ph} \frac{\partial \mu}{\partial S_{ph}} (S_{ph}^{(i)}, S_{cr}^{(i)}) + k_{cr} \frac{\partial \mu}{\partial S_{cr}} (S_{ph}^{(i)}, S_{cr}^{(i)}) \right) \right],$$

$$i = 2, 3.$$

The third root $\lambda_3^{(i)} = -X^{(i)} \left( k_{ph} \frac{\partial \mu}{\partial S_{ph}} (S_{ph}^{(i)}, S_{cr}^{(i)}) + k_{cr} \frac{\partial \mu}{\partial S_{cr}} (S_{ph}^{(i)}, S_{cr}^{(i)}) \right)$ is found numerically by computing the right-hand side expression on a discrete mesh of values for $D$, where $D \in (D_{cr}^{(1)}, D_{cr})$ for $E_2$, and $D \in (D_{cr}^{(1)}, D_{cr}^{(2)})$ for $E_3$. Figure 3 visualizes the three eigenvalues of $J(E_2)$ and $J(E_3)$. One can see that the eigenvalues of $J(E_3)$ are negative (right plot), and $J(E_2)$ possesses one real positive eigenvalue (left plot). Moreover, one eigenvalue of $J(E_2)$ approaches zero at $D = D_{cr}^{(1)}$, and one eigenvalue of $J(E_3)$ approaches zero at $D = D_{cr}^{(1)}$ and $D = D_{cr}^{(2)}$, thus $D_{cr}^{(1)}$ and $D_{cr}^{(2)}$ are bifurcation parameter values.

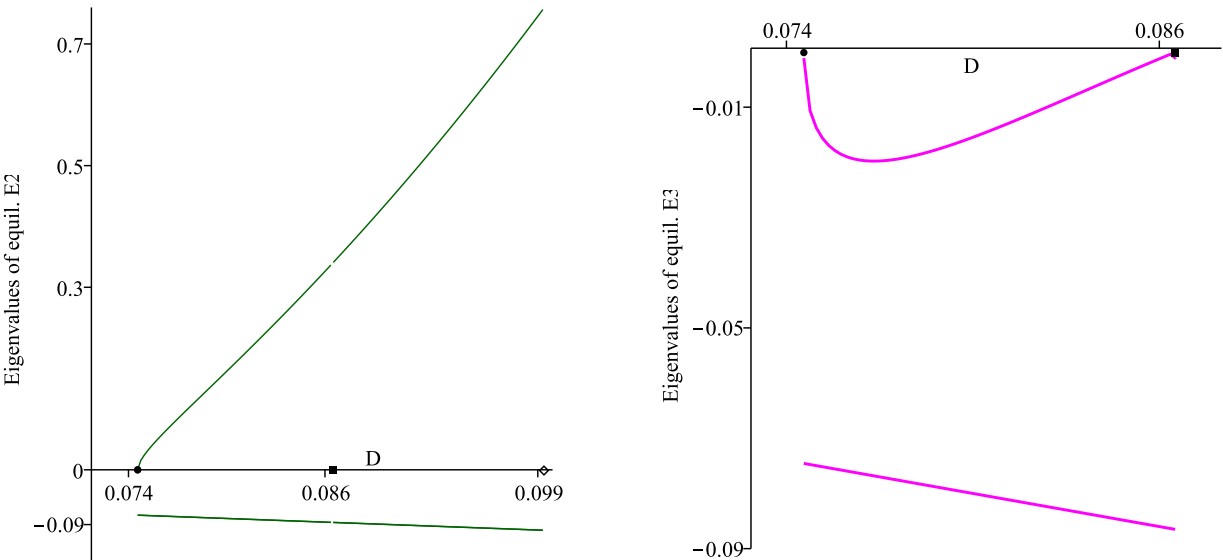

**Figure 3.** Eigenvalues corresponding to the equilibrium points $E_2$ (**left**) and $E_3$ (**right**), parameterized on $D$. On the horizontal axis, the solid circle denotes $D_{cr}^{(1)}$, the solid box denotes $D_{cr}^{(2)}$, the diamond denotes $D_{cr}$.

We summarize the above results in the next proposition.

**Proposition 2.**

(i)    *The equilibrium $E_2$, defined for $D \in (D_{cr}^{(1)}, D_{cr})$, is locally asymptotically unstable (a saddle).*

(ii)    *The equilibrium $E_3$, defined for $D \in (D_{cr}^{(1)}, D_{cr}^{(2)})$, is locally asymptotically stable (a stable node).*

(iii)    *At $D = D_{cr}^{(1)}$, the two interior equilibrium points, $E_2$ and $E_3$, are 'born', thus $D_{cr}^{(1)}$ is a bifurcation value of the parameter $D$. At $D = D_{cr}^{(1)}$ the steady states $E_2$ and $E_3$ undergo a saddle-node bifurcation.*

(iv)    *At $D = D_{cr}^{(2)}$ the equilibrium points $E_3$ and $E_0$ coalesce and exchange stability for $D > D_{cr}^{(2)}$. Thus, at $D = D_{cr}^{(2)}$ the steady states $E_3$ and $E_0$ undergo a transcritical bifurcation.*

Figure 1 (right plot) and Figure 2 also visualize the stability of $E_0$, $E_2$ and $E_3$: the solid lines correspond to the components of the stable equilibria, the dash and the dash-dot lines mark the components of the unstable equilibria. Therefore, these three plots can also be considered as bifurcation diagrams: a saddle node bifurcation occurs at the parameter value $D = D_{cr}^{(1)}$, and $D = D_{cr}^{(2)}$ serves as a transcritical bifurcation point.

## 5. Global Stabilizability of the Model Dynamics

First we prove that the model (1)–(3) exhibits the standard properties that we would expect from a bioreactor model, namely uniqueness and positiveness of solutions for non-negative initial conditions.

**Theorem 1.** *Consider the model (1)–(3) and assume that $X(0) \geq 0$, $S_{ph}(0) \geq 0$, $S_{cr}(0)) \geq 0$.*

(i)   *If $X(0) = 0$ then all model solutions tend to the equilibrium point $E_0 = (0, S_{ph}^0, S_{cr}^0)$.*

(ii)   *If $X(0) > 0$ then $X(t) > 0$, $S_{ph}(t) > 0$, $S_{cr}(t) > 0$ for all $t > 0$.*

(iii)   *All solutions are uniformly bounded for all $t \geq 0$.*

**Proof.** (i) Let $X(0) = 0$ and $S_{ph}(0) \geq 0$, $S_{cr}(0) \geq 0$ be satisfied. It follows that $X(t) = 0$ for all $t \geq 0$ due to uniqueness of solutions of the Cauchy problem. Then the model (1)–(3) reduces to

$$\frac{dS_{ph}(t)}{dt} = D(S_{ph}^0 - S_{ph}(t))$$

$$\frac{dS_{cr}(t)}{dt} = D(S_{cr}^0 - S_{cr}(t)).$$

The latter equations imply that $S_{ph}(t)$ and $S_{cr}(t)$ converge exponentially to $S_{ph}^0$ and $S_{cr}^0$ respectively. The plane $X = 0$ is invariant for the model.

(ii)–(iii) Assume that $X(0) > 0$, $S_{ph}(0) \geq 0$, $S_{cr}(0)) \geq 0$. It follows from Equation (1) that

$$\frac{dX}{X} = \int_0^t (\mu(S_{ph}(\tau), S_{cr}(\tau)) - D)d\tau,$$

$$X(t) = X(0)e^{\int_0^t (\mu(S_{ph}(\tau), S_{cr}(\tau)) - D)d\tau} > 0 \text{ for each } t \geq 0.$$

Denote $\Sigma_1(t) = S_{ph}(t) + k_{ph}X(t) - S_{ph}^0$. Then Equations (1) and (2) imply

$$\frac{d}{dt}\Sigma_1(t) = \frac{dS_{ph}}{dt} + k_{ph}\frac{dX}{dt} = D\left(S_{ph}^0 - (S_{ph} + k_{ph}X)\right) = -D\Sigma_1(t),$$

which means that $\Sigma_1(t) = e^{-Dt}\Sigma_1(0)$, thus $\lim_{t \to \infty} \Sigma_1(t) = 0$, or equivalently

$$\lim_{t \to \infty} \left(S_{ph}(t) + k_{ph}X(t)\right) = S_{ph}^0.$$

Since $X(t) > 0$ for all $t > 0$ this means $S_{ph}(t) > 0$ for all $t > 0$ as well. Moreover, $X(t)$ and $S_{ph}(t)$ are uniformly bounded.

Similarly, using Equations (1) and (3) and denoting $\Sigma_2(t) = S_{cr}(t) + k_{cr}X(t) - S_{cr}^0$ we obtain $\Sigma_2(t) = e^{-Dt}\Sigma_2(0)$, which means that

$$\lim_{t \to \infty} (S_{cr}(t) + k_{cr}X(t)) = S_{cr}^0. \tag{16}$$

Therefore, $S_{cr}(t) > 0$ for all $t > 0$ and $S_{cr}(t)$ is uniformly bounded for $t \geq 0$. Hence, the model solutions $X(t)$, $S_{ph}(t)$, $S_{cr}(t)$ exist for all time $t \geq 0$. This completes the proof of Theorem 1.   □

In the following we shall prove the global asymptotic stabilizability of system (1)–(3) when the control parameter $D$ belongs to the interval $\left( D_{cr}^{(1)}, D_{cr}^{(2)} \right)$, with $D_{cr}^{(2)} = \mu(S_{ph}^0, S_{cr}^0)$.

Similarly to the proof of Theorem 1, denote $\Sigma_3(t) = S_{ph}(t) - KS_{cr}(t) - S^0$, where $K$ and $S^0$ are defined in (9). After multiplying Equation (3) by $-\dfrac{k_{ph}}{k_{cr}}$ and adding to Equation (2) we obtain

$$
\begin{aligned}
\frac{d}{dt}\Sigma_3(t) &= \frac{d}{dt}\left( S_{ph}(t) - KS_{cr}(t) \right) = D\left( S_{ph}^0 - S_{ph}(t) - KS_{cr}^0 + KS_{cr}(t) \right) \\
&= D\left( (S_{ph}^0 - KS_{cr}^0) - (S_{ph}(t) - KS_{cr}(t)) \right) \\
&= -D\left( S^0 - (S_{ph}(t) - KS_{cr}(t)) \right) = -D\Sigma_3(t).
\end{aligned}
$$

This means that $\Sigma_3(t) = e^{-Dt}\Sigma_3(0)$, $\Sigma_3(0) \geq 0$, so $\lim_{t\to\infty}\Sigma_3(t) = 0$. Then system (1)–(3) may be written in the form

$$
\begin{aligned}
\frac{d}{dt}\Sigma_3(t) &= -D\Sigma_3(t) \\
\frac{d}{dt}X(t) &= \left( \mu(S^0 + KS_{cr}(t), S_{cr}(t)) - D \right)X(t) \\
\frac{d}{dt}S_{cr}(t) &= -k_{cr}\mu(S^0 + KS_{cr}(t), S_{cr}(t))X(t) + D(S_{cr}^0 - S_{cr}(t)).
\end{aligned}
$$

Since $\lim_{t\to\infty}\Sigma_3(t) = 0$, the positive $\omega$-limit set of any solution of system (1)–(3) is contained in the set

$$
\Omega_3 = \left\{ (X, S_{ph}, S_{cr}) : \ X > 0, S_{ph} \geq 0, S_{cr} \geq 0, \Sigma_3 = 0 \right\}.
$$

Using the theory of the asymptotically autonomous systems (cf. [20,21]) it follows that all trajectories forming the $\omega$-limit set of any solution of (1)–(3) with initial conditions in $\Omega_3$ are solutions of the following limiting system

$$
\begin{aligned}
\frac{dX(t)}{dt} &= \left( \mu(S^0 + KS_{cr}(t), S_{cr}(t)) - D \right)X(t) \\
\frac{dS_{cr}(t)}{dt} &= -k_{cr}\,\mu(S^0 + KS_{cr}(t), S_{cr}(t))X(t) + D(S_{cr}^0 - S_{cr}(t)).
\end{aligned}
\tag{17}
$$

We consider Equation (17) on the set

$$
\Omega_2 = \left\{ (X, S_{cr}) : \ X > 0, S_{cr} \geq 0, S^0 + KS_{cr} \geq 0 \right\}.
$$

Denote for simplicity $\mu_{cr}(S_{cr}) = \mu(S^0 + KS_{cr}, S_{cr})$. Then obviously $\mu_{cr}(S_{cr}^0) = \mu(S^0 + KS_{cr}^0, S_{cr}^0) = \mu(S_{ph}^0, S_{cr}^0) = D_{cr}^{(2)}$ holds true.

Let us choose an arbitrary value $\bar{D} \in \left( D_{cr}^{(1)}, D_{cr}^{(2)} \right)$, and consider the following system obtained from (17) after substituting $D = \bar{D}$ in the latter:

$$
\frac{dX(t)}{dt} = \left( \mu_{cr}(S_{cr}(t)) - \bar{D} \right)X(t)
\tag{18}
$$

$$
\frac{dS_{cr}(t)}{dt} = -k_{cr}\,\mu_{cr}(S_{cr}(t))X(t) + \bar{D}(S_{cr}^0 - S_{cr}(t)).
\tag{19}
$$

Let us recall, that at $D = \bar{D}$ there are two interior equilibria of the model (1)–(3),

$$E_2(\bar{D}) = (X^{(2)}(\bar{D}), S_{ph}^{(2)}(\bar{D}), S_{cr}^{(2)}(\bar{D})) \quad \text{and}$$

$$E_3(\bar{D}) = (X^{(3)}(\bar{D}), S_{ph}^{(3)}(\bar{D}), S_{cr}^{(3)}(\bar{D})) \quad \text{with} \quad S_{cr}^{(2)}(\bar{D}) < S_{cr}^{(3)}(\bar{D}).$$

Denote

$$\bar{E} = (\bar{X}, \bar{S}_{cr}) = \left( X^{(3)}(\bar{D}), S_{cr}^{(3)}(\bar{D}) \right).$$

Obviously, $\bar{E}$ is an equilibrium point of (18) and (19).
We make the following assumption.

**Assumption 1.** *There exist points $S_{cr}^-$ and $S_{cr}^+$ such that $0 < S_{cr}^- < S_{cr}^+ < S_{cr}^0$ and $\mu_{cr}(S_{cr})$ is monotone increasing for all $S_{cr} \in (S_{cr}^-, S_{cr}^+)$.*

Assumption 1 identifies the equilibrium $\bar{E}$ with the projection of $E_3(\bar{D})$ in the plane $S_{ph} - KS_{cr} = S^0$. If we choose for $S_{cr}^-$ the $S_{cr}$-component of the double root of Equation (11), i.e., $S_{cr}^- = S_{cr}^{(2)}(D_{cr}^{(1)}) = S_{cr}^{(3)}(D_{cr}^{(1)})$, and $S_{cr}^+ = S_{cr}^0$, then $\mu_{cr}(S_{cr})$ is monotone increasing in $(S_{cr}^-, S_{cr}^+)$, see the left plot in Figure 1.

Based on the above considerations, the problem for global stabilizability of the model (1)–(3) is reduced to proving the global stabilizability of the well known basic bioreactor (chemostat) model (17), which is well studied in the literature, see e.g., [20,22–24] and the references therein. The next Theorem 2 is also a corollary from Theorem 2.1 in [25]. We present the proof here for reader's convenience.

**Theorem 2.** *Let Assumption 1 be fulfilled. Assume that $\bar{D} \in \left( D_{cr}^{(1)}, D_{cr}^{(2)} \right)$. Then for any initial point $(X(0), S_{cr}(0)) \in \Omega_2$ the corresponding solution of (18) and (19) converges asymptotically towards the equilibrium point $\bar{E}$.*

**Proof.** Let us fix an arbitrary initial point $(X(0), S_{cr}(0)) \in \Omega_2$.

First we shall show that there exists time $T > 0$, such that $S_{cr}(t) < S_{cr}^0$ for all $t > T$. Assume that $S_{cr}(t) \geq S_{cr}^0$ holds true for each $t > 0$. Then we have from (19) that

$$\frac{dS_{cr}(t)}{dt} = -k_{cr}\,\mu_{cr}(S_{cr})X(t) + \bar{D}(S_{cr}^0 - S_{cr}(t)) < 0.$$

Barbălat's Lemma [26] implies

$$0 = \lim_{t \to \infty} \frac{dS_{cr}(t)}{dt} = \lim_{t \to \infty} \left( -k_{cr}\,\mu_{cr}(S_{cr})X(t) + \bar{D}(S_{cr}^0 - S_{cr}(t)) \right),$$

which leads to $S_{cr}(t) \to S_{cr}^0$ and $X(t) \to 0$ as $t \to \infty$. Further we have that $\mu_{cr}(\bar{S}_{cr}) = \bar{D} < D_{cr}^{(2)} = \mu_{cr}(S_{cr}^0)$. The continuity of $\mu_{cr}(\cdot)$ and the relation $S_{cr}(t) \to S_{cr}^0$ as $t \to \infty$ imply that there exists a number $\delta > 0$ such that

$$\mu_{cr}(S_{cr}(t)) - \bar{D} = \mu_{cr}(S_{cr}(t)) - \mu_{cr}(\bar{S}_{cr}) \geq \delta$$

for all sufficiently large $t$. Then it follows $\dfrac{dX(t)}{dt} = (\mu_{cr}(S_{cr}(t)) - \bar{D})X(t) \geq \delta X(t)$ for all sufficiently large $t$, which contradicts the boundedness of $X(t)$. Hence, there exists a sufficiently large $T > 0$ with $S_{cr}(T) \leq S_{cr}^0$. If the equality $S_{cr}(T) = S_{cr}^0$ holds true, then we have

$$\frac{dS_{cr}}{dt}(T) = -k_{cr}\,\mu_{cr}(S_{cr}(T))X(T) + \bar{D}(S_{cr}^0 - S_{cr}(T))$$
$$= -k_{cr}\,\mu_{cr}(S_{cr}(T))X(T) < 0.$$

The last inequality shows that $S_{cr}(t) < S_{cr}^0$ for each $t > T$.

Let us fix an arbitrary $\gamma \in \left(0, (\mu_{cr}(S_{cr}^0) - \mu_{cr}(\bar{S}_{cr}))/2\right)$. (Note that $\mu_{cr}(S_{cr})$ is monotone increasing.) The continuity of $\mu_{cr}$ implies that there exists $\varepsilon > 0$ such that $\mu_{cr}(\bar{S}_{cr}) + \gamma < \mu_{cr}(S_{cr})$ for each $S_{cr} \in \left[S_{cr}^0 - (1 + k_{cr})\varepsilon, S_{cr}^0\right)$. It follows from (16) that there exists time $T_\varepsilon > 0$ so that $X(t)$ and $S_{cr}(t)$ satisfy

$$S_{cr}^0 - \varepsilon < S_{cr}(t) + k_{cr}X(t) < S_{cr}^0 + \varepsilon \quad \text{for each } \ t \geq T_\varepsilon. \tag{20}$$

Assume now that $X(\bar{t}) \leq \varepsilon$ for some $\bar{t} \geq T_\varepsilon$; then we obtain from (20)

$$S_{cr}^0 > S_{cr}(\bar{t}) \geq S_{cr}^0 - k_{cr}X(\bar{t}) - \varepsilon \geq S_{cr}^0 - (1 + k_{cr})\varepsilon,$$

i.e., $S_{cr}(\bar{t}) \in \left[S_{cr}^0 - (1 + k_{cr})\varepsilon, S_{cr}^0\right)$. Hence,

$$\frac{d}{dt}X(\bar{t}) = (\mu_{cr}(S_{cr}(\bar{t})) - \bar{D})X(\bar{t}) = (\mu_{cr}(S_{cr}(\bar{t})) - \mu_{cr}(\bar{S}_{cr}))X(\bar{t}) \geq \gamma X(\bar{t}) > 0.$$

It follows then that $X(t) \geq e^{(t-\bar{t})\gamma}X(\bar{t})$. If there exists $t_1 \geq \bar{t}$ such that $X(t_1) = \varepsilon$, then at every time $t_2 \geq t_1$ with $X(t_2) = \varepsilon$ we have $\frac{d}{dt}X(t_2) = (\mu_{cr}(S_{cr}(t_2)) - \bar{D})X(t_2) \geq \gamma\varepsilon > 0$. Hence there exists time $T_1 > T$ such that $X(t) \geq \varepsilon$ for each $t \geq T_1$.

The above considerations mean that the $\omega$-limit set of the corresponding trajectory of (18) and (19) lies in the set

$$\{(X, S_{cr}) : \ X \geq \varepsilon, \ 0 \leq S_{cr} \leq S_{cr}^0\}.$$

For $X > 0$ and $S_{cr} \in (0, S_{cr}^0)$ we define the following Lyapunov function

$$V = V(X, S_{cr}) = \int_{\bar{X}}^{X} \frac{\eta - \bar{X}}{\eta}d\eta + \int_{\bar{S}_{cr}}^{S_{cr}} \frac{\bar{X}(\mu_{cr}(\xi) - \bar{D})}{\bar{D}(S_{cr}^0 - \xi)}d\xi.$$

The derivative $\dfrac{d}{dt}V$ of $V$ along the solutions of (18) and (19) is presented by

$$\begin{aligned}
\frac{d}{dt}V &= \frac{X - \bar{X}}{X}(\mu_{cr}(S_{cr}) - \bar{D})X + \frac{\bar{X}(\mu_{cr}(S_{cr}) - \bar{D})}{\bar{D}(S_{cr}^0 - S_{cr})}\left(-k_{cr}\mu_{cr}(S_{cr})X + \bar{D}(S_{cr}^0 - S_{cr})\right)\\
&= X(\mu_{cr}(S_{cr}) - \bar{D})\left(1 - \frac{\bar{X}k_{cr}\mu_{cr}(S_{cr})}{\bar{D}(S_{cr}^0 - S_{cr})}\right)\\
&= X(\mu_{cr}(S_{cr}) - \bar{D})\left(1 - \frac{S_{cr}^0 - \bar{S}_{cr}}{S_{cr}^0 - S_{cr}} \cdot \frac{\mu_{cr}(S_{cr})}{\bar{D}}\right) \leq 0
\end{aligned}$$

for each $S_{cr} \in (0, S_{cr}^0)$ and $X > 0$. Applying LaSalle's invariance principle it follows that each trajectory of (18) and (19) approaches the equilibrium point $\bar{E}$, i.e., $\bar{E}$ is globally asymptotically stable. This proves the theorem. □

It follows from Propositions 1 and 2 that when the control input $D$ takes values $D > D_{cr}^{(2)} = \mu(S_{ph}^0, S_{cr}^0)$ then the model (1)–(3) possesses two equilibrium points—the wash-out equilibrium $E_0 = (0, S_{ph}^0, S_{cr}^0)$ and the interior equilibrium $E_2$, such that $E_0$ is locally asymptotically stable and $E_2$ is locally asymptotically unstable. Using the reduced model (17) it can be shown that the restriction $\bar{E}_0 = (0, S_{cr}^0)$ of the wash-out equilibrium $E_0$ is globally asymptotically stable if $D > D_{cr}^{(2)} = \mu_{cr}(S_{cr}^0)$. Although the proof can be extracted from the more general Lemma 2.2 in [24], we present it below for completeness.

**Theorem 3.** *Assume that $D > D_{cr}^{(2)}$ holds true. Then for any initial point $(X(0), S_{cr}(0)) > 0$ the corresponding solution of (17) converges asymptotically towards the equilibrium $\bar{E}_0 = (0, S_{cr}^0)$.*

**Proof.** Choose some $\bar{D}_0 > D_{cr}^{(2)}$ and consider system (18) and (19), where $\bar{D}$ is replaced by $\bar{D}_0$. Assume that $\lim_{t\to\infty} X(t) = X^* > 0$. Then Barbălat's Lemma [26] applied to Equation (18) implies $0 = \lim_{t\to\infty} \frac{d}{dt} X(t) = \lim_{t\to\infty} (\mu_{cr}(S_{cr}(t)) - \bar{D}_0)X^*$, which means that $\lim_{t\to\infty} \mu_{cr}(S_{cr}(t)) = \bar{D}_0 > \mu_{cr}(S_{cr}^0)$. From the continuity of $\mu_{cr}(\cdot)$ it follows that there exists time $T > 0$ and a positive number $\delta$ such that $\mu_{cr}(S_{cr}(t)) - \mu_{cr}(S_{cr}^0) \geq \delta$ for all $t \geq T$. The latter inequality leads to $\frac{d}{dt} X(t) \geq \delta X(t)$, a contradiction with the boundedness of $X(t)$. Therefore, $X(t) \to 0$ as $t \to \infty$. From the theory of the asymptotically autonomous systems (cf. [20,21]) it follows that the dynamics (18) and (19) can be reduced to the limiting equation $\frac{d}{dt} S_{cr}(t) = \bar{D}_0(S_{cr}^0 - S_{cr}(t))$, which implies $\lim_{t\to\infty} S_{cr}(t) = S_{cr}^0$, and this completes the proof. □

## 6. Dynamic Behavior of the Model Solutions: Numerical Simulation

In this section we present two numerical examples that illustrate the dynamic behavior of the model solutions.

**Example 1.** $D = 0.08 \in (D_{cr}^{(1)}, D_{cr}^{(2)})$

In this case there exist two positive (coexistence) equilibrium points

$$E_2 = (0.0491, 0.126, 0.0153) \text{ and } E_3 = (0.0318, 0.328, 0.116),$$

such that $E_2$ is locally asymptotically unstable, $E_3$ is the globally asymptotically stable equilibrium point according to Theorem 2. The wash-out equilibrium $E_0 = (0, 0.7, 0.3)$ is locally asymptotically unstable.

The left plot in Figure 4 visualizes the convergence of the solutions towards the corresponding equilibrium components of $E_3$ using two different starting points. The right plot of Figure 4 as well as Figure 5 visualize projections of the trajectories in the phase planes $(X, S_{cr})$, $(X, S_{ph})$ and $(S_{ph}, S_{cr})$ respectively with three different initial points, denoted by circles. The corresponding projections of the invariant planes are marked by dash lines in the three plots.

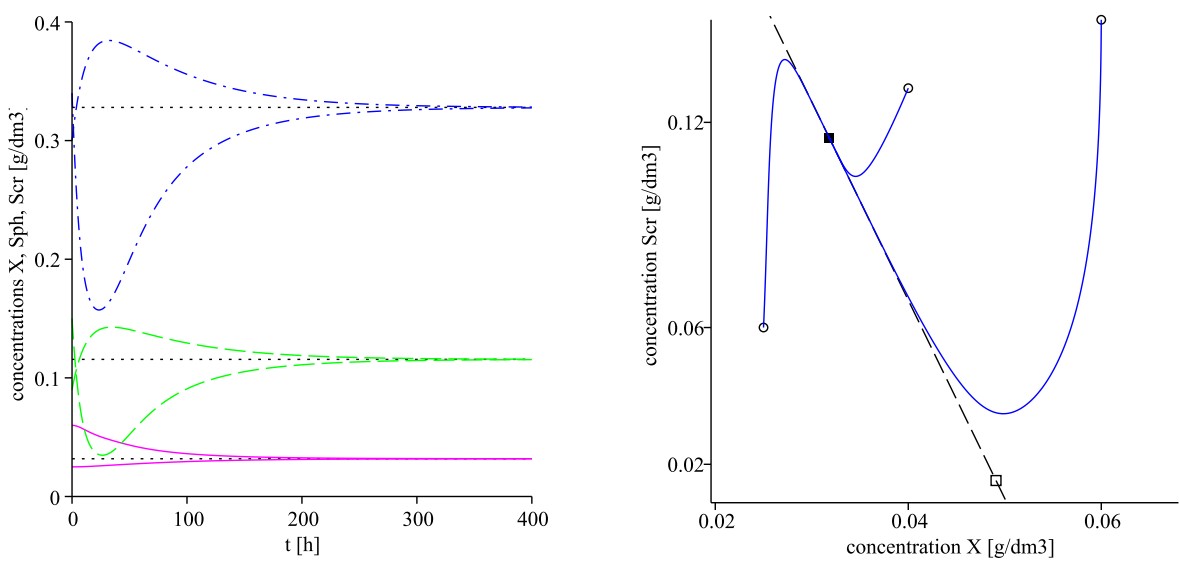

**Figure 4.** $D = 0.08$. (**Left**): time evolution of solutions $X$ (solid line), $S_{ph}$ (dash-dot line) and $S_{cr}$ (dash line); the horizontal dot lines pass through the corresponding components of the equilibrium point $E_3$. (**Right**): projections of the trajectories in the $(X, S_{cr})$-phase plane with three different initial points, denoted by circles. The corresponding equilibrium components of $E_3$ are marked by a solid box, of $E_2$ are denoted by a box. The dash line presents the a projection of invariant plane $S_{cr} + k_{cr}X = S_{cr}^0$.

**Example 2.** $D = 0.095 > D_{cr}^{(2)} \approx 0.0865$

In this case there exists only one interior equilibrium point $E_2 = (0.0514, 0.0987, 0.00191)$ which is locally asymptotically unstable. The wash-out equilibrium $E_0 = (0, 0.7, 0.3)$ is the globally asymptotically stable steady state according to Theorem 3.

The global stability of $E_0$ for large values of the control parameter $D$ means total wash-out of the biomass $X$ and thus no detoxification of the bioreactor medium.

The left plot in Figure 6 visualizes the convergence of the solutions towards the corresponding components of $E_0$ using two different starting points. The right plot of Figure 6 as well as Figure 7 visualize projections of the trajectories in the phase planes $(X, S_{cr})$, $(X, S_{ph})$ and $(S_{ph}, S_{cr})$ respectively with three different initial points, marked by circles. The latter three plots also visualize the projections of the invariant planes in the corresponding phase planes.

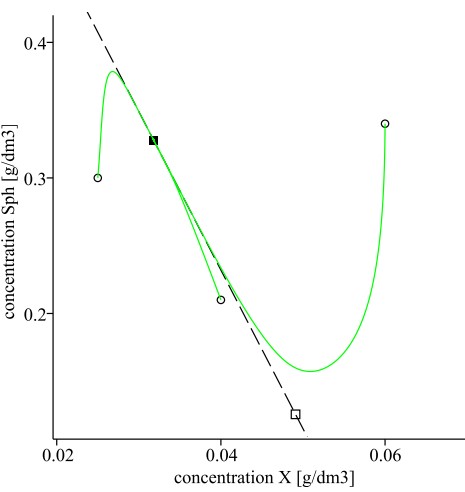 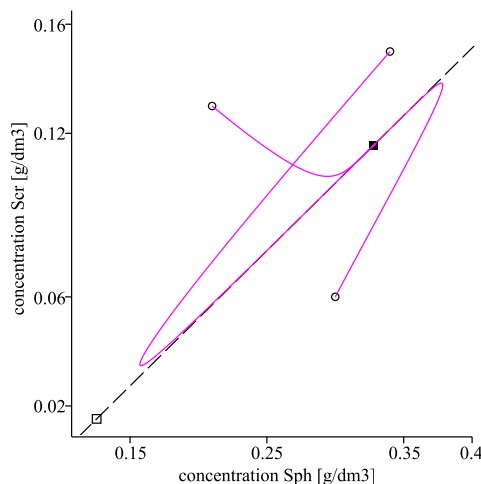

**Figure 5.** $D = 0.08$. Projections of the trajectories in the $(X, S_{ph})$-phase plane (**left**) and in the $(S_{ph}, S_{cr})$-phase plane (**right**) with three different initial points, denoted by circles. The corresponding equilibrium components of $E_3$ are marked by solid boxes, of $E_2$ are denoted by boxes. The dash lines present projections of the invariant planes $S_{ph} + k_{ph}X = S_{ph}^0$ (left) and $S_{ph} - KS_{cr} = S^0$ (right).

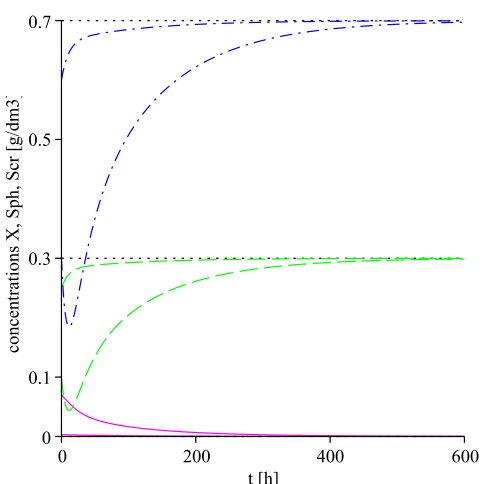 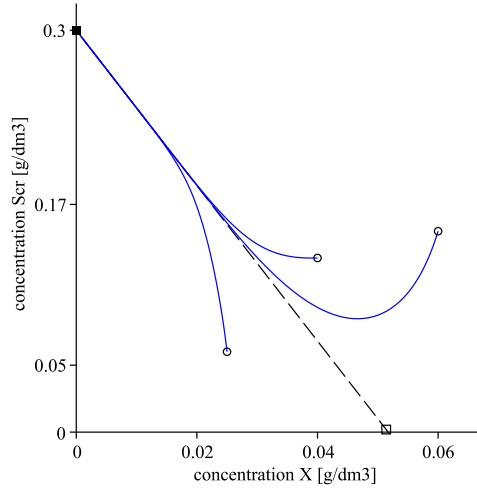

**Figure 6.** $D = 0.095$. (**Left**): time evolution of solutions $X$ (solid line), $S_{ph}$ (dash-dot line) and $S_{cr}$ (dash line); the horizontal dot lines pass through the corresponding components of the equilibrium point $E_0$. (**Right**): projections of the trajectories in the $(X, S_{cr})$-phase plane with three different initial points, denoted by circles. The corresponding equilibrium components of $E_0$ are marked by a solid box, of $E_2$ are denoted by a box. The dash line presents a projection of the invariant plane $S_{cr} + k_{cr}X = S_{cr}^0$.

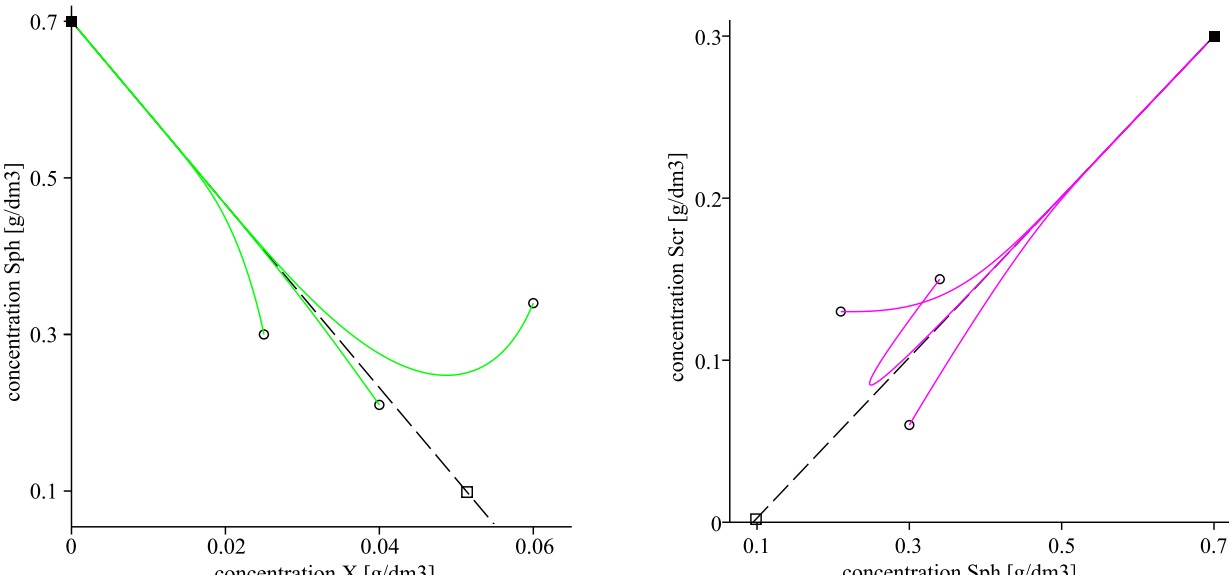

**Figure 7.** $D = 0.095$. Projections of the trajectories in the $(X, S_{ph})$-phase plane (**left**) and in the $(S_{ph}, S_{cr})$-phase plane (**right**) with three different initial points, denoted by circles. The corresponding components of $E_0$ are marked by solid boxes, of $E_2$ are denoted by boxes. The dash lines present projections of the invariant planes $S_{ph} + k_{ph}X = S_{ph}^0$ (left) and $S_{ph} - KS_{cr} = S^0$ (**right**).

## 7. Conclusions

We perform a mathematical analysis of a dynamic model, describing phenol and 4-methylphenol (*p*-cresol) biodegradation in a continuously stirred tank bioreactor. The model is described by three nonlinear ordinary differential equations and presents an extension of the batch growth model given in [17] to perform the ability of *Aspergillus awamori* strain to degrade the mixture of phenol and *p*-cresol. The novel idea is the usage of *sum kinetic interaction parameters* in the analytic expression of the microorganisms specific growth rate $\mu(S_{ph}, S_{cr})$ in the medium, as well as inhibition terms with respect to both phenol and *p*-cresol concentrations. The advantages of using such kind of specific growth rates is validated by practical laboratory experiments [17], see also [5]. To our knowledge, such kind of dynamic models, describing biodegradation in continuous biorectors (chemostats), are not studied in the literature until now.

We compute the equilibrium points of the model and investigate their local asymptotic stability as well as existence of bifurcations in dependence of the input control parameter $D$, the dilution rate. It is shown that an equilibrium $E_0 = \left(0, S_{ph}^0, S_{cr}^0\right)$, corresponding to total wash-out of the biomass in the bioreactor, exists for all $D > 0$. We find values of $D$ such that two interior (coexistence) equilibria $E_2$ and $E_3$ do exist: $E_3$ is defined for $D \in \left(D_{cr}^{(1)}, D_{cr}^{(2)}\right)$, and $E_2$ exists if $D \in \left(D_{cr}^{(1)}, D_{cr}\right)$. Local stability analysis shows that $E_2$ is locally asymptotically unstable and $E_3$ is locally asymptotically stable where they exist, $E_0$ is locally asymptotically stable for $D > D_{cr}^{(2)}$. Two types of bifurcations of the equilibria occur, a saddle node bifurcation at $D = D_{cr}^{(1)}$ where $E_2$ and $E_3$ coalesce, and a transcritical bifurcation at $D = D_{cr}^{(2)}$, where $E_0$ coincides with $E_3$ and $E_3$ disappears for $D > D_{cr}^{(2)}$. Practically, the bifurcation values $D_{cr}^{(1)}$ and $D_{cr}^{(2)}$ of $D$ should be carefully avoided, because small nearby perturbations may cause destabilization of the process, leading to total wash-out of the biomass. Most of the computations are carried out numerically due to the complicated expression of the model function $\mu(\cdot)$ and the large number of model parameters. The computations are performed in the computer algebra system *Maple*. The most important property of the model solutions—existence, uniqueness and uniform boundedness—is established theoretically in Theorem 1. We also prove (Theorem 2) the

global asymptotic stability of the interior equilibrium point $E_3$ when $D$ takes values within certain bounds, $D \in \left( D_{cr}^{(1)}, D_{cr}^{(2)} \right)$, $D_{cr}^{(2)} = \mu(S_{ph}^0, S_{cr}^0)$. The existence of these bounds for $D$ is not restrictive in practical applications, since the dilution rate $D$ is proportional to the speed of the pumping mechanism which feeds the bioreactor, thus there always exist a lower and an upper bound for $D$ [27]. Choosing $D$ in the interval $\left( D_{cr}^{(1)}, D_{cr}^{(2)} \right)$ ensures practically long-term sustainability of the bioremediation process in the bioreactor. On the other hand, large values of the dilution rate $D$, $D > D_{cr}^{(2)} = \mu(S_{ph}^0, S_{cr}^0)$, may cause total wash-out of the biomass in the reactor and may lead to process breakdown. This is due to the fact that the wash-out equilibrium $E_0 = (0, S_{ph}^0, S_{cr^0})$ is the global attractor of the dynamics (Theorem 3). The dynamic behavior of the model solutions is illustrated by some numerical examples for different values of the dilution rate.

**Author Contributions:** Conceptualization, N.D. and P.Z.; methodology, N.D.; software, N.D.; theoretical investigation, N.D.; validation, N.D. and P.Z.; data curation, P.Z.; writing—original draft preparation, N.D.; writing—review and editing, N.D. and P.Z. Both authors have read and agreed to the published version of the manuscript.

**Funding:** This research received no external funding.

**Acknowledgments:** This work has been partially supported by the National Scientific Program "Information and Communication Technologies for a Single Digital Market in Science, Education and Security (ICTinSES)", contract No DO1–205/23.11.2018, financed by the Ministry of Education and Science in Bulgaria. The work of the first author has been partially supported by grant No BG05M2OP001-1.001-0003, financed by the Science and Education for Smart Growth Operational Program (2014–2020) in Bulgaria and co-financed by the European Union through the European Structural and Investment Funds.

**Conflicts of Interest:** The authors declare no conflict of interest.

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
