# Peer review of "Global Stability Analysis of a Bioreactor Model for Phenol and Cresol Mixture Degradation"

_processes, doi:10.3390/pr9010124_

Round 1

Reviewer 1 Report

The present manuscript describes the mathematical analysis, concerning the global stability, of a dynamic model for two compounds biodegradation on a continuously stirred tank bioreactor with a very interesting application. The manuscript is very well structured and written. Some minor comments, concerning the references: some journal names are not abbreviated and some doi are missing. The introduction section could have more detailed examples.

Reviewer 2 Report

The authors present and analyse a model for the biodegradation of two substrates in a bio-reactor system. In particular, they demonstrate that the system has stable equilibria, identify ranges in which those equilibria occur, and the types of bifurcation in the system. The analysis follows standard methods and is essentially correct.

The authors state that the key novelty of their results is the use of "sum kinetics with interaction parameters" (SKIP). I am willing to accept their statement that this is an innovation within bio-reactor modelling, but I would note that models of this type are extensively used in ecological modelling. Specifically, this is fundamentally a model of a predator with two prey and some cost to hunting both simultaneously. I would like to see some reference to that existing literature. Additionally, I would like to see more justification that this additional model complexity is needed in this application.   

Reviewer 3 Report

The submitted article “Global stability analysis of a bioreactor model for phenol and cresol mixture degradation” proposes a mathematical model in which the authors used the specific growth rate representing the kinetics with interaction parameters with the inhibition terms for cell growth on phenol and p-cresol. They investigated the behavior of the model by finding the steady states, illuminating their existence and local stability properties. Lastly, the authors show global stability property of the interior equilibrium and give numerical examples visualized in MAPLE.

The presented manuscript is, in my view, written well. The presentation and structure are clear. I have not had enough time to evaluate the correctness of all the derivations, but from what I have reviewed it seems that there are no problems either. The topic of the manuscript fits well within the journal Processes and therefore I recommend it to be published.
